# A_2_BC-Type Porphyrin SAM on Gold Surface for Bacteria Detection Applications: Synthesis and Surface Functionalization

**DOI:** 10.3390/ma14081934

**Published:** 2021-04-13

**Authors:** Laurie Neumann, Lea Könemund, Valentina Rohnacher, Annemarie Pucci, Hans-Hermann Johannes, Wolfgang Kowalsky

**Affiliations:** 1Labor für Elektrooptik/Institut für Hochfrequenztechnik, TU Braunschweig, Bienroder Weg 94/Schleinitzstraße 22, D-38106 Braunschweig, Germany; laurie.neumann@ihf.tu-bs.de (L.N.); lea.koenemund@ihf.tu-bs.de (L.K.); wolfgang.kowalsky@ihf.tu-bs.de (W.K.); 2Kirchhoff Institute for Physics, Heidelberg University, Im Neuenheimer Feld 227, D-69120 Heidelberg, Germany; valentina.rohnacher@kip.uni-heidelberg.de (V.R.); pucci@kip.uni-heidelberg.de (A.P.)

**Keywords:** porphyrin, SAM formation, modified gold surface, *E. coli* immobilization

## Abstract

Currently used elaborate technologies for the detection of bacteria can be improved in regard to their time consumption, labor intensity, accuracy and reproducibility. Well-known electrical measurement methods might connect highly sensitive sensing systems with biological requirements. The development of modified sensor surfaces with self-assembled monolayers (SAMs) from functionalized porphyrin for bacteria trapping can lead to a highly sensitive sensor for bacteria detection. Different A_2_BC-type porphyrin structures were synthesized and examined regarding their optical behavior. We achieved the synthesis of a porphyrin for SAM formation on a gold surface as electrode material. Two possible bio linkers were attached on the opposite *meso*-position of the porphyrin, which allows the porphyrin to react as a linker on the surface for bacteria trapping. Different porphyrin structures were attached to a gold surface, the SAM formation and the respective coverage was investigated.

## 1. Introduction

New materials for integration into sensor surfaces for bacteria detections are of great interest in medicine, food safety, public health and security. Due to the high mortality caused by bacterial infection, fast and reliable measurement methods are essential [1]. Conventional bacterial detection methods like cell culture, microscopic analysis and biochemical assays have long processing times, insufficient sensitivity and specificity, and require specialized equipment and educated personnel, which make these methods costly and not always available [2]. So far two different methods for sensors are described for bacterial detection. One method requires sample preparation for the detection of bacterial components such as nucleic acids (DNA and RNA) or exotoxins, this leads to longer measurement times and higher costs [3,4,5]. The other method detects the complete bacterial cell without further sample preparation. For whole bacterial cell detection optical, mechanical or electrochemical sensor systems are investigated which are functionalized on their electrode surface with different bioreceptors like antibodies, artificial binding proteins, DNA/RNA aptamers or bacteriophages [6,7,8]. Bio-inspired porphyrins as a modifier for these sensing systems are of interest due to their biological compatibility, their fluorescence in the visible range, and their chemical stability under ambient conditions. Another advantage is their simple chemical accessibility. So far porphyrins have already been used in several sensor systems for example for magnesium (II) ion, histidine or DNA detection [9,10,11]. They can be used as chemical sensors for gaseous analytes, for liquid phase analytes as electrochemical, photoelectrochemical or optical sensors [12,13,14,15]. In one example from Kawata et al., a graphene field-effect transistor functionalized with a tetrakis-(4-carboxyphenyl) porphyrin as the linker demonstrated highly sensitive biosensing [16].

We synthesized modified porphyrin structures for self-assembled monolayer (SAM) formation on a gold surface. These porphyrin SAMs have the possibility to act as a bacteria trapping antenna in biosensing systems and might result in high sensitivity and a fast bacteria detection in aqueous solution. For future bacteria detection this functionalized gold electrode will be integrated as the gate electrode in a modfied field-effect transistor where the bacteria trapping might lead to changes in the electric potential. For SAM formation on the gate electrode, different 5,15-A_2_BC-type porphyrins were synthesized originating from a 5,15-diphenylporphyrin with the possibility to connect to a gold surface through a thioacetyl group. In the following synthesis steps an additional ligand was attached with a peptide functionality for Gram-negative bacteria trapping. It has already been shown that this peptide can bind to the outer membrane of Gram-negative *E. coli* (*Escherichia coli)* [17]. So far porphyrins with the thioacetyl group have been applied to a gold surface with a modified SAM formation process in solution [18]. This porphyrin linker showed its ability of SAM formation on the gold electrode surface and was proven by methods like ultraviolet–visible (UV–vis) spectroscopy, DSA (drop-shape-analysis), IRRAS (infrared reflection-absorption spectroscopy) and CV (cyclic voltammetry). In addition, the SAM functionalized gold substrates were examined regarding their ability of Gram-negative bacteria linkage with the method of FLIM (fast lifetime imaging microscopy) measurements. For the bacteria linkage we used 4′,6-diamidino-2-phenylindole (DAPI)—stained *E. coli* K12. This verification is the first step for the development of a new porphyrin modified surface for the linkage and detection of Gram-negative bacteria.

## 2. Materials and Methods

### 2.1. General Measurements

^1^H nuclear magnetic resonance (NMR) and ^13^C NMR spectra were recorded using a AVIII400, AVIIIHD500 and AVII600 from Bruker (Billerica, MA, USA). Chemical shifts are reported in ppm units with tetramethylsilane as an internal standard. Electrospray ionization mass spectrometry (ESI-MS) measurements were performed on a Finnigan 8400-MSS I and MAT 4515 (Bremen, Germany). UV–vis spectra were measured on a Perkin-Elmer Lambda 9 (Waltham, MA, USA) at room temperature in the range of 300–800 nm. For the measurements in solution the molar extinction coefficient ε is indicated in L mol^−1^∙cm^−1^. For the gold substrate measurements, a gold reference substrate was used to obtain the background spectrum for the functionalized substrate absorption and the substrates were measured in reflection. The photoluminescence spectra were measured on a Cary Eclipse spectrometer from Agilent Technologies (Santa Clara, CA, USA) with a resolution of 1 nm. Contact angle measurements were performed on a Krüss Drop Shape Analyzer DSA25 (Hamburg, Germany) with a tilt stage with the Krüss Advance Software (Version 1.3.1.0, Hamburg, Germany). The free surface energy was measured on a sessile drop with deionized water and diiodomethane for 10 contact angles on two gold substrates. The analysis was calculated with the method of Owens, Wendt, Rabel and Kaelble (OWRK model). Cyclic voltammetry (CV) was performed on a Bio-Logic Science Instruments GmbH MPG-2 (Seyssinet-Pariset, France) with 50 mV∙s^−1^ at a potential range of −100 mV to 500 mV. For porphyrin **12** the measurement range was extended to −200 mV and 700 mV in order to make the redox reactions visible. Three scans were recorded and averaged for the CV discussions. As reference electrode a Ag/AgCl double-junction electrode from Deutsche Metrohm GmbH and Co KG (Filderstadt, Germany) filled with a 3 mol/L KCl solution as inner filling and a 0.1 mol/L KCl solution as outer filling was chosen. A gold thin film substrate with 3 nm chromium and 100 nm gold was used as a counter electrode. The gold substrates were chosen as working electrode in an aqueous solution with 1 mmol/L K_4_[Fe(CN)_6_] as the analyte and 0.1 mmol/L KCl as supporting electrolyte. Crocodile clips fixed the counter and working electrodes during measurements. It was ensured that the area of the counter electrode immersed in the electrolyte has at least the same size as the working electrode. The IRRAS measurements were performed using a Bruker Vertex 80v FTIR (Fourier transform infrared, Ettlingen, Germany) spectrometer equipped with a liquid nitrogen-cooled mercury cadmium telluride detector from MCT. The samples were measured with a motorized stage in the sample compartment at 3 mbar. The spectra were obtained using p-polarized light and are averaged over 1000 scans with a resolution of 2 cm^−1^. The raw spectra of the molecules on gold were divided by a reference spectrum of a clean gold substrate, thus giving the relative reflectance R of the molecular layer. For the FLIM measurements a MicroTime 100 (MT100) setup from Picoquant (Berlin, Germany) was used. As the laser source a LDH-P-C375B for an excitation at 375 nm was used. A confocal microscope BX43 from Olympus was equipped with a MPlanFLN objective with 100× magnification and the surface was focused in transmission before FLIM measurements. The SymPhoTime 64 software (Version 2.4, Berlin, Germany) was used for FLIM data acquisition and an image size of 465 pixels was measured.

### 2.2. Self-Assembled Monolayer (SAM) Film Preparation

Gold substrates were produced in an electron beam evaporator on 25 mm × 25 mm borosilicate glass substrates from Schott with a thickness of 0.7 mm, which were previously washed with acetone und isopropanol in an ultrasonic bath. After washing, the borosilicate glass substrates were coated with 3 nm chromium as adhesion promoter and 100 nm gold layer. For the FLIM measurements glass substrates with a thickness of 175 µm ± 15 µm from Paul Marienfeld GmbH and Co. KG were used and instead of 100 nm, a 20 nm gold layer was promoted. Prior to use, the gold substrates were activated in an oxygen plasma for 3 min. The porphyrin SAMs were prepared by an acid-promoted method [18], where the appropriate porphyrin (2 ×∙10^−4^ mol/L) was dissolved in a solvent mixture of CH_2_Cl_2_ (18 mL) and MeOH (8 mL). After adding 420 μL of sulfuric acid the porphyrin solution was stirred for 2 h at room temperature for deprotection of the thiol moiety. Under argon atmosphere in a glove box, four gold substrates were added to a petri dish in a sealable polypropylene vessel and the porphyrin solution was added. The gold substrates were immersed at room temperature for 3 days, followed by washing with a CH_2_Cl_2_/MeOH (2/1, (*v/v*)) solution in an ultrasonic bath for 5 min and rinsed with a CH_2_Cl_2_/MeOH solution afterward. This process was repeated. Finally, the substrates were washed with CH_2_Cl_2_/MeOH solution and dried for 1 h at 70 °C under high vacuum or at 80 °C under normal pressure over night.

### 2.3. Bacteria Culture, Staining and Substrate Incubation

For bacteria immobilisation on the gold substrates the *E. coli* K12 strain was used. It was cultured in lysogeny-broth (LB) medium. A single colony was added to 50 mL of liquid LB medium and incubated at 37 °C on a shaker incubator (150 rpm) overnight followed by a subculture until an OD_600 nm_ (optical density at 600 nm) of 0.5–0.8 was reached. For cell harvesting, 20 mL of the bacterial suspension was centrifuged. Sedimented cells were washed three times with sterile phosphate-buffered saline (PBS, pH 7.2) and finally resuspended in PBS to a final OD_600 nm_ of 0.6–0.8. The bacteria cells were then treated with 20 mL of 300 nmol/L DAPI stain solution in PBS and incubated for 10 min at 37 °C in the dark on a shaker incubater (150 rpm). Afterwards the cells were washed with PBS to remove the unbound DAPI and were finally resuspended in PBS before substrate incubation. After that, the reference and the functionalized gold substrates were fully covered with the bacterial suspension in a centrifuge tube and incubated for 10 min at 37 °C in the dark on a shaker incubater (150 rpm). Then, the substrates were transferred to a centrifuge tube with PBS and incubated for 3 min, shaked on a vortexer within this time and finally rinsed with PBS. After the washing process, the substrates were placed on a microscope slide, wetted with PBS and a coverslip was added before FLIM measurements.

### 2.4. Synthesis

All chemicals and solvents were purchased from Acros Organics (Fair Lawn, NJ, USA), Sigma Aldrich (St. Louis, MO, USA) and Tokyo Chemical Industries (Tokyo, Japan). The peptide was purchased from Genscript Biotech (Piscataway Township, NJ, USA). All syntheses were performed under nitrogen atmosphere in anhydrous solvents.


**4-((4-(10,20-diphenylporphyrin-5-yl)phenyl)ethynyl)benzyl)ethanethioate (Porphyrin 5)**


To a solution of anhydrous toluene (400 mL) and triethylamine (120 mL), porphyrin **3** (200 mg, 0.36 mmol) and 1-(*S*-acethylthiomethyl)-4-iodbenzol (**4**, 540 mg, 1.85 mmol) were added. The solution was flushed with nitrogen for 10 min. After adding of AsPh_3_ (132 mg, 0.43 mmol) and Pd_2_(dba)_3_ (49 mg, 0.05 mmol) the solution was stirred for 48 h at 35 °C. The solvent was evaporated and the residue was purified using a flash silica gel column (4/3, (*v/v*) CH_2_Cl_2_/hexane) to receive the product as a purple solid (170 mg, 65%).

^1^H-NMR (600 MHz, CDCl_3_) *δ* = –3.10 (br s, 2H, NH), 2.30 (s, 3H), 4.10 (s, 2H), 7.2 (d, *J* = 8.0 Hz, 2H), 7.50 (d, *J* = 8.0 Hz, 2H) 7.70 (m, 6H), 7.80 (d, *J* = 7.9 Hz, 2H), 8.10 (d, *J* = 8.0 Hz, 2H), 8.10 (d, *J* = 7.6 Hz, 4H), 8.8 (d, *J* = 4.7 Hz, 2H), 8.8 (d, *J* = 4.7 Hz, 2H), 8.9 (d, *J* = 4.5 Hz, 2H), 9.2 (d, *J* = 4.5 Hz, 2H), 10.1 (s, 1H) ppm. ESI-MS: Calcd. for C_49_H_34_N_4_OS, 726.25; found (M + H^+^), 727.25. UV–vis (CH_2_Cl_2_): λ_max_ (log ε) 413 (5.55), 509 (4.14), 542 (3.69), 583 (3.54), 638 (3.20) nm.


**4-((4-(15-bromo-10,20-diphenylporphyrin-5-yl)phenyl)ethynyl)benzyl)ethanethioate (Porphyrin 6)**


A solution of porphyrin **5** (150 mg, 0.21 mmol) in CHCl_3_ (48 mL) and MeOH (6 mL) was cooled down to 0 °C. *N*-bromosuccinimide (41 mg, 0.23 mmol) was added to the solution, stirred for 10 min at 0 °C and was then allowed to warm up to room temperature. After stirring overnight, the reaction was quenched with acetone (10 mL). The solvent was evaporated and the residue was purified using a flash silica gel column (1/1, (*v/v*), CH_2_Cl_2_/hexane) to receive the product as a purple solid (125 mg, 74%).

^1^H-NMR (500 MHz, CDCl_3_) *δ* = −2.84 (br s, 2H, NH), 2.30 (s, 3H), 4.07 (s, 2H), 7.25 (d, J = 8.2 Hz, 2H), 7.50 (d, J = 8.2 Hz, 2H), 7.65–7.72 (m, 6H), 7.80 (d, J = 8.3 Hz, 2H), 8.07 (d, J = 8.3 Hz, 2H), 8.09–8.10 (m, 4H), 8.72 (s, 4H), 8.80 (d, J = 4.7 Hz, 2H), 9.57 (d, J = 4.7 Hz, 2H) ppm. ESI-MS: Calcd. for C_49_H_33_BrN_4_OS, 804.16; found (M^+^), 804.16. UV–vis (CH_2_Cl_2_): λ_max_ (log ε) = 422 (5.65), 519 (4.22), 554 (4.02), 596 (3.66), 652 (3.68) nm.


**4-((4-(15-(4-nitrophenyl)-10,20-diphenylporphyrin-5-yl)phenyl)ethynyl)benzyl) ethanethioate (Porphyrin 7)**


K_3_PO_4_ (1.30 g, 6.20 mmol) was dried prior to use, followed by adding anhydrous THF (100 mL). Afterwards porphyrin **6** (200 mg, 0.25 mmol), 4(4,4,5,5-tetramethyl-1,3,2-dioxaborolan-2-yl)anilin (742 mg, 2.98 mmol) and Pd(PPh_3_)_4_ (81 mg, 0.07 mmol) were added. The reaction was stirred for 7 h at 85 °C under the exclusion of light. The solvent was then evaporated and the residue redissolved in CH_2_Cl_2_, washed with sat. NaHCO_3(aq)_, dist. H_2_O and NaCl_(aq)_. The organic phase was dried over Na_2_SO_4_ and the solvent was evaporated. The residue was purified using a flash silica gel column (CH_2_Cl_2_) to receive the product as a purple solid (178 mg, 83%).

^1^H-NMR (600 MHz, CDCl_3_) *δ* = −2.86 (br s, 2H, NH), 2.30 (s, 3H), 4.06 (s, 2H), 7.25 (d, *J* = 8.3 Hz, 2H), 7.50 (d, *J* = 8.2 Hz, 2H), 7.64–7.71 (m, 6H), 7.80 (d, *J* = 8.3 Hz, 2H), 8.09 (d, *J* = 8.2 Hz, 2H), 8.10–8.12 (m, 4H), 8.26 (d, *J* = 8.6 Hz, 4H), 8.49 (d, *J* = 8.6 Hz, 2H), 8.63 (d, *J* = 4.6 Hz, 2H), 8.77–8.79 (m, 4H), 8.80 (d, *J* = 4.6 Hz, 2H) ppm. ESI-MS: Calcd. for C_55_H_37_N_5_O_3_S, 847.26; found (M^+^), 847.26. UV–vis (CH_2_Cl_2_): λ_max_ (ε) = 420 (5.36), 516 (4.35), 552 (4.23), 591 (4.12), 647 (4.07) nm.


**4-((4-(15-(4-aminophenyl)-10,20-diphenylporphyrin-5-yl)phenyl)ethynyl)benzyl)ethanethioate (Porphyrin 8)**


Porphyrin **7** (130 mg, 0.15 mmol) was dissolved in concentrated HCl (11 mL) and Sn(II)Cl_2_ (162 mg, 0.86 mmol) was slowly added. The reaction was stirred for 3 h under nitrogen at 65 °C. After cooling down, the solution was poured into dist. H_2_O and was adjusted to pH 8 with an aqueous ammonium hydroxide solution. The product was extracted with CH_2_Cl_2_, the organic solvent was dried over Na_2_SO_4_ and the solvent was evaporated. The residue was purified using a flash silica gel column (6 /1.5, (*v/v*), CH_2_Cl_2_/EA) and after drying under high vacuum the product was received as a shiny purple solid (93 mg, 66%). ^1^H-NMR (500 MHz, CDCl_3_) *δ* = –2.83 (br s, 2H, NH), 2.29 (s, 3H), 3.88 (br s, 2H, NH), 4.06 (s, 2H), 6.91 (d, *J* = 8.2 Hz, 2H), 7.26 (d, *J* = 8.0 Hz, 2H), 7.51 (d, *J* = 8.0 Hz, 2H), 7.63–7.68, m, 6H), 7.80–7.82 (m, 2H), 7.88 (d, *J* = 8.2 Hz, 2H), 8.10–8.13 (m, 6H), 8.74–8.76 (m, 6H), 8.85–8.86 (m, 2H) ppm. ESI-MS: Calcd. for C_55_H_39_N_5_OS, 817.29; found (M + H^+^), 818.29. UV–vis (CH_2_Cl_2_): λ_max_ (log ε) = 423 (5.32), 519 (4.18), 556 (4.03), 591 (3.88), 650 (3.81) nm.


**(Z)-4-((4-(15-(4-((4-((acetylthio)methyl)phenyl)ethynyl)phenyl)-10,20-diphenylporphyrin-5-yl)phenyl)amino)-4-oxobut-2-enoic acid (Porphyrin 9)**


Maleic anhydride (58 mg, 0.59 mmol) was added in portions to a solution of porphyrin **8** (97 mg, 0.11 mmol) in anhydrous THF (30 mL). The solution was stirred over night at room temperature and was then poured into a 0.1 mol/L NaOH-solution (100 mL). The product was extracted with CH_2_Cl_2_, the organic solvent was dried over Na_2_SO_4_ and the solvent was evaporated. The residue was purified using a silica gel column (3/1, (*v/v*), CH_2_Cl_2_/MeOH) to receive the product as a red-purple solid (96 mg, 87%).

ESI-MS: Calcd. for C_59_H_41_N_5_O_4_S, 915.29; found (M + H^+^), 916.29. UV–vis (dimethyl sulfoxide (DMSO)): λ_max_ (log ε) = 423 (5.38), 518 (4.08), 555 (3.92), 593 (3.68), 649 (3.65) nm.


**S-(4-((4-(15-(4-(2,5-dioxo-2,5-dihydro-1H-pyrrol-1-yl)phenyl)-10,20-diphenylporphyrin-5-yl)phenyl)ethynyl)benzyl) ethanethioate (Porphyrin 10)**


Porphyrin **9** (40 mg, 0.04 mmol) was dissolved in acetic anhydride (2.5 mL) and sodium acetate (7 mg, 0.1 mmol) was added. After stirring for 3 h at 100 °C, the reaction was quenched with dist. H_2_O (10 mL). The product was extracted with CH_2_Cl_2_. The organic solvent was dried over Na_2_SO_4_ and the solvent was evaporated. The residue was purified using a silica gel column (CH_2_Cl_2_) to receive the product as a purple solid. (29 mg, 75%). ^1^H-NMR (400 MHz, CDCl_3_) *δ* = −2.86 (br s, 2H, NH), 2.29 (s, 3H), 4.06 (s, 2H), 6.86 (s, 2H), 7.22–7.25 (m, 2H), 7.49–7.51 (d, *J* = 8.2 Hz, 1H), 7.56–7.58 (d, *J* = 8.2 Hz, 1H), 7.64–7.68 (m, 8H), 7.80–7.85 (m, 2H), 8.10–8.13 (m, 6H), 8.20–8.22 (m, 2H), 8.78–8.79 (m, 8) ppm. ESI-MS: Calcd. for C_59_H_39_N_5_O_3_S, 897.28; found (M + H^+^), 898.28. UV–vis (DMSO): λ_max_ (log ε) = 421 (5.48), 516 (4.11), 552 (3.83), 591 (3.63), 646 (3.55) nm.


**S-(1-(4-(15-(4-((4-((acetylthio)methyl)phenyl)ethynyl)phenyl)-10,20-diphenylporphyrin-5-yl)phenyl)-2,5-dioxopyrrolidin-3-yl)-N-amino-L-isoleucyl-L-phenylalanyl-L-cysteinyl-L-phenylalanyl-L-lysyl-L-arginyl-L-lysyl-L-arginyl-L-lysil-L-tryptophyl-L-leucyl-L-valyl-L-tyrosine (Porphyrin 11)**


Porphyrin **10** (6 mg, 0.007 mmol) and the peptide YVLWKRKRKFCFI-amide (12 mg, 0.007 mmol) were dissolved in DMSO (2 mL) and *N,N*-diisopropylethylamine (0.01 mL) in DMSO (0.1 mL) was added. The solution was stirred at room temperature for 48 h. The solvent was then given in Et_2_O (100 mL), filtered and redissolved in CH_2_Cl_2_/MeOH (1/1, (*v/v*)). The solvent was evaporated and the product received as a red-brown solid (12 mg, 61%).

ESI-MS: Calcd. for C_147_H_174_N_29_O_17_S_2_, 2681.31; found (M/2z^+^ + 2H) 1342.67, (M/3z^+^ + 2H) 895.45, (M/4z^+^ + H) 671.84. UV–vis (DMSO): λ_max_ (log ε) = 421 (5.47), 516 (4.14), 552 (3.90), 591 (3.70), 647 (3.65) nm.


**S-(1-(4-(15-(4-((4-((acetylthio)methyl)phenyl)ethynyl)phenyl)-10,20-diphenylporphyrin-5-yl)phenyl)-2,5-dioxopyrrolidin-3-yl)-l-cysteine (Porphyrin 12)**


Porphyrin **10** (6 mg, 0.007 mmol) and l-cysteine (0.9 mg, 0.007 mmol) were dissolved in DMSO (2 mL) and the solution was stirred for 48 h at room temperature. The solvent was then given in Et_2_O (100 mL), filtered and redissolved in CH_2_Cl_2_/MeOH (1/1, (*v/v*)). The solvent was evaporated and the product received as a red solid (8 mg, 100%).

ESI-MS: Calcd. for C_62_H_46_N_6_O_5_S_2_, 1018.30; found (M + H^+^), 1019.30. UV–vis (DMSO): λ_max_ (log ε) = 422 (5.21), 517 (3.90), 553 (3.69), 592 (3.48), 647 (3.44) nm.

## 3. Results and Discussion

### 3.1. Synthesis of A_2_BC-Type Porphyrins and Photophysical Properties

Following the synthetic pathway (**Scheme 1**) different A_2_BC-type porphyrins are synthesized and characterized by NMR spectroscopy, ESI-MS and UV–vis absorption spectroscopy. Dipyrromethane (**1**), porphyrin **2** and **3** and 1-(S-acethylthiomethyl)-4-iodobenzene (**4**) were obtained according to literature procedures [19,20,21,22]. The porphyrins **5***–***10** were synthesized according to literature with minor changes to the procedures [20,21,23,24,25,26]. For further biological applications, porphyrin **10** was modified with a peptide with the sequence YVLWKRKRKFCFI-amide, for porphyrin **11** or cysteine as the amino acid to yield porphyrin **12**. It has already been shown that this peptide can be connected to the outer membrane of Gram-negative bacteria due to its high compatibility to neutralize lipopolysaccharide [17,27]. For further studies on surface functionalization porphyrin **5** and porphyrin **10***–***12** were used as the SAMs.

The UV–vis absorption and the photoluminescence results (**Table 1**) show the SAM porphyrins for the surface functionalization and were measured in solvent in comparison to the starting A_2_B_2_-type porphyrin **2**. For all porphyrins in solution a very intense *Soret* band between 407–422 nm and four relatively weak Q bands between 502–649 nm can be observed. For the photoluminescence (PL) measurements an excitation wavelength of 420 nm was chosen for all porphyrins, measured in dimethyl sulfoxide. The absorption and the emission spectra show a bathochromic shift with additional ligands in *meso*-position of the porphyrins. With the *meso*-substituted aryl-ligands for porphyrin **5** and porphyrin **10** the electron-donating effect influences the absorption and emission towards a red shift of the wavelength compared to porphyrin **2** [28]. After peptide or the cysteine group was added to the porphyrin **10**, no significant shifts in absorption or emission can be observed, the peptide—as well as the cysteine—group have no further influence on the optical behavior of the porphyrin.

### 3.2. SAM Formation on Gold Substrates

#### 3.2.1. Absorption Properties

The absorption spectra of the porphyrin SAMs on gold in comparison to the porphyrins in dimethyl sulfoxide are shown in **Figure 1**. For all porphyrins a broadening and a red-shift of the *Soret* band are observed. Porphyrin **5** shows a red shift of 23 nm as SAM on gold, porphyrin **10** a 27 nm red shift, porphyrin **11** a 22 nm red shift and porphyrin **12** similar with a red shift of 21 nm. Due to the orientation of the porphyrin SAMs on the surface a red shift or blue shift of the wavelength can occur. If face-to-face aggregation of the porphyrins on the surface occurs, a blue shift of the wavelength is observed, in comparison, a red shifted wavelength is the result of edge-to-edge aggregation [29,30,31]. It is also reported, that a broadening of the *Soret* band results from a mixture of J- and H-aggregates [32]. For porphyrin **5** a stronger broadening in comparison to porphyrin **10** can be observed in **Figure 1**, which shows that we have a more uniform SAM formation for porphyrin **10**. Porphyrin **12** with the cysteine group on gold results in a less distinct broadening. The comparison of the porphyrin **11** with the porphyrin **12** functionalized gold substrate shows that we observe a lower uniform SAM formation for the additionally peptide group. One reason for this might be the sterically hindered peptide chain which caused a hindered SAM formation. Our UV–vis study shows that the porphyrins are demonstrably on the surface. Since the porphyrin SAMs are shifted to longer wavelength (red shift), a predominant side-by-side orientation of the porphyrins can be concluded [18,32].

#### 3.2.2. Contact Angle

For chemical changes of the surface due to SAM formation on the gold electrode, contact angle measurements were used to determine information about the hydrophobicity and/or hydrophilicity, due to chemical changes [33]. For surface free energy measurements diiodomethane and water were used. The contact angles (**Table 2**) were measured on pure gold substrates and gold substrates functionalized with porphyrin **5** and porphyrins **10–12**. In comparison to the gold reference a variation of the polarity behavior of the surface can be observed.

The gold substrate functionalized with porphyrin **5** as monolayer shows an increasing hydrophobicity of the surface in comparison to the gold reference. This provides information on the orientation, where the porphyrin molecules are stacked next to each other with the nonpolar side of the ring is aligned towards the top. For porphyrin **10** and **11** on gold an increasing of the hydrophilicity on the surface can be observed, as the porphyrin is modified with the polar maleimide group for porphyrin **10** and the peptide group for porphyrin **11**. The cysteine-bearing porphyrin **12** shows no significant changes in comparison to the gold reference surface, this might indicate that the cysteine groups of porphyrin **12** interact through the carboxyl group which leads to a surface saturation. The surface free energy of porphyrin SAM gold substrates in relation to the gold reference substrate are illustrated and are between a range of 51.55 mN/m–55.15 mN/m. Due to their amino acid groups for porphyrin **11** and **12**, it is expected to show a similar or more polar surface in comparison to porphyrin **10** with the two carbonyl groups of the maleimide group. Nevertheless, the results show a lower polarity of the surface. One reason for this could be a strong interaction between the amino acid groups of the porphyrins, so that the carbonyl groups are not oriented to the top. Another possible reason is that the surface has larger gaps between the porphyrins due to their sterical demand created by the peptide group of porphyrin **11** and the cysteine group of porphyrin **12**.

#### 3.2.3. Cyclic Voltammetry

With the method of cyclic voltammetry metal electrodes can be analyzed with respect to their electron transfer efficiency and surface coverage [33]. Here K_4_[Fe(CN)_6_] was chosen as analyte for its sensitivity to surface changes. The blank gold electrode was used as the working electrode and the typical redox potentials are presented (**Figure 2a**). The effect of a hindered electron transport due to SAM formation can be observed for porphyrin **5** and porphyrin **10** because of the missing redox peaks compared to the reference sample (**Figure 2b**). Nevertheless, the anodic and cathodic current increases as the applied potential becomes increasingly positive or negative. Electrons can transfer on three different ways between the electroactive species and the electrode surface. (1) The electroactive species can diffuse to defects in the monolayer or (2) permeate through the functional layer and react with the electrode surface. (3) Electrons can also transfer through the monolayer via a tunneling process over the molecules [33,35]. Since the redox peaks compared to the reference sample are missing, options (1) and (2) are not dominant for porphyrin **5** and **10**. The increasing anodic and cathodic current on the edge of the measuring range supposes the diffusion of the electroactive species to regions with a sparsely coverage of the functional layer which facilitate the tunneling process [35]. The maximum anodic current for porphyrin **10** is about a factor 0.66 smaller compared to porphyrin **5**. Porter et al. reported similar results about monolayer assemblies on *n*-alkyl thiols with different chain lengths on gold layers. Ellipsometry, infrared spectroscopy and electrochemical results proved higher film organization relating to longer chains due to distinct van der Waals interactions [35]. Here, it can be assumed that the maleimide group on porphyrin **10** improves the interaction between adjacent molecules. Due to the carbonyl group of the maleimide group, hydrogen bonds besides van der Waals interactions are formed. The IRRAS results supports this assumption because the C=O stretching vibration at 1670 cm^−1^ is missing (**Figure 3**). The film seems to be higher organized which prohibits the tunneling process. This aspect is even in good agreement with the UV–vis results shown in **Figure 1**. The absorption peak of porphyrin **10** is narrower compared to porphyrin **5.** Higher film organization leads to a higher formation of J-aggregates and the absence of H-aggregates.

The cyclic voltammograms of porphyrin **11** and **12** show oxidation and reduction peaks. The measurement range was increased for porphyrin **12** in order to make the redox reactions visible. Porphyrin **12** and **10** differ only in the cysteine attached on the maleimide group. According to the IRRAS results the C=O stretching vibration is still missing. Besides the carbonyl group on the maleimid group the carboxyl group of the cysteine is involved in hydrogen bonds. Stronger intermolecular interactions are supposed compared to porphyrin **10**. This hypothesis is in good agreement with the UV–vis results shown in **Figure 1**. The narrow absorption peak can be traced back to a higher organized SAM with porphyrin **12** compared to porphyrin **10**. Nevertheless, the cathodic and anodic current differs clearly (**Figure 2c**). Qingwen et al. documented the redox behavior of cysteine as monolayer on gold surfaces with electrochemical techniques. Compared to blank gold surfaces cysteine modified gold layers showed an improvement of the reversibility of the redox reaction and higher peak currents. The observation is traced back to the existence of a positive charged surface. The –NH_2_ group of cysteine may carry a positive charge in the described PBS solution which improves the reduction of the analyte [36]. This consideration may also be the case in the shown cyclic voltammograms for porphyrin **12** and can be an explanation of the measured oxidation and reduction processes. In contrast to the results presented by Qingwen et al., the peak current did not rise and the reversibility of the redox reaction did not improve. The additional porphyrin structure besides the cysteine enables but does not favor the charge transfer compared to the reference sample.

Porphyrin **11** shows an even narrower potential difference between the anodic and cathodic peak and higher peak currents compared to the reference sample (**Figure 2d**). As similarly explained in the context of porphyrin **12**, the peptide might carry positive charges when immersed in the electrolyte during measurement. But as already discussed in the context of porphyrin **12,** amino acids enable but do not favor the charge transfer. Therefore, there has to be a second electron transfer option to explain the cyclic voltammogram for porphyrin **11**. As shown in **Scheme 1** the long peptide chain from porphyrin **11** is complex and increases the width of the molecule significantly. As already discussed in the contact angle discussion, steric repulsion might lead to a large mean distance between adjacent molecules in the porphyrin SAM. Permeation of the analyte in the functional layer and the subsequent diffusion to the electrode surface for redox reactions is facilitated and contributes the measured cyclic voltammogram.

#### 3.2.4. Infrared Reflection Absorption Spectroscopy

Infrared reflection-absorption spectroscopy (IRRAS) was used to prove the formation of the monolayers of the different A_2_BC-type porphyrins. The IRRA spectra of different A_2_BC-type porphyrins (**Figure 3**) shows characteristic vibrations of the molecules. The overall low intensity of IR absorption bands of about <0.2% indicate the formation of a monolayer of porphyrin molecules on the gold substrate. The strong mode in the blue spectra of porphyrin **11** at 1500 cm^−1^–1700 cm^−1^ in the grey marked region can be assigned to the peptide chain which is attached to the porphyrin molecule (**Scheme 1**). Amide bonds within the peptide chain absorb radiation in multiple regions of the IR spectrum, including a strong band at around 1670 cm^−1^ which originates from the C=O stretching vibration of the peptide bond [37,38]. As this strong band cannot be observed for porphyrin **12** with the cysteine group, this can be seen as an additional indication of an interaction of the carboxylic groups between the porphyrin within the SAM formation, as already discussed. The specific mode at 1520 cm^−1^ results from the N–H bending vibration and from the C–N stretching vibration [37,38,39]. The red IRRA spectra (**Figure 3**) of porphyrin **5** contains at 800 cm^−1^ the specific mode of the C–H bending vibration of the porphyrin ring, which is visible in all four IRRA spectra of the different A_2_BC-type porphyrins. The black IRRA spectra of porphyrin **10** shows a new mode at 960 cm^−1^ which can be assigned to the newly added headgroup (**Scheme 1**). With the IRRAS measurements we could also demonstrate, besides the UV–vis, contact angle und CV measurements, the presence of the porphyrins on the gold surface.

### 3.3. Fast Lifetime Imaging Microscopy (FLIM)

FLIM measurements are used for detection of lifetime properties of the fluorescent DAPI stained *E. coli* cells on the gold substrate surface. FLIM has several advantages, as it can be used to detect changes in the molecular environments of fluorophores [40]. With the stained bacteria cells with a specific lifetime, it can be ensured that the structures shown are the stained *E. coli* cells located on the surface and not possible defects on the gold surface. **Figure 4A** shows the FLIM image of a single DAPI stained *E. coli* cell on a microscope slide at an excitation of 375 nm. After bacteria treatment on the gold reference substrate in **Figure 4B** it can be observed that there are almost no bacterial cells, as there is just one spot which shows the DAPI stained *E. coli* lifetime from **Figure 4A**. Whereas for the porphyrin **12** and porphyrin **11** functionalized gold substrates in **Figure 4C,D** a large number of lifetime spots can be observed. It can be concluded that this shows the attached bacteria cell distribution on the substrate surface. With both porphyrin structures it is possible to connect Gram-negative *E. coli* cells to a gold surface. Noticeable is the higher number of bacterial cells on the porphyrin **12** functionalized surface with the cysteine group in comparison to the porphyrin **11** functionalized surface with the peptide group. One reason for this could be a more uniform SAM formation on the gold surface for porphyrin **12** in comparison to porphyrin **11** with the sterically demanding peptide group, as the *E. coli* cell has more binding opportunities with a higher number of porphyrins on the gold surface.

## 4. Conclusions

We synthesized different A_2_BC-type porphyrins with a thioacetyl group in *meso*-position for SAM formation on a gold surface. Four different porphyrins were investigated, whereas two of them are attached with a promising bacteria binding group. For all porphyrins, a SAM formation on gold and a mainly side-by-side orientation can be demonstrated with UV–vis spectroscopy. Detailed discussions about the SAM formation and interaction between the adjacent modified porphyrins could be undertaken from the cyclic voltammetry results. With the help of IRRAS measurements, it was possible to show the formation of a monolayer of the porphyrins **5** and **10**–**12**, and differences in the IRRA spectra can be visualized because of the different ligand groups of these porphyrins. Due to the changes of the polarity of the surface within the drop shape analysis, orientation of the bacteria trapping group along to the top of the gold substrate can be demonstrated. We synthesized and integrated successfully a porphyrin structure on a gold surface which has the ability to be used for bacteria trapping. In a next step it was also possible to show the ability to act as a whole-cell *E. coli* trapper for porphyrin **11** and **12** on a gold surface. As we can successfully connect the Gram-negative bacteria through the porphyrin on the gold surface, these structures are a promising candidate for bacteria-sensing applications. Further investigations into the porphyrin SAM distribution and their orientation on the gold surface are of interest.

## Data Availability

The data presented in this study are available on request from the corresponding author.

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
