# Peer review of "A2BC-Type Porphyrin SAM on Gold Surface for Bacteria Detection Applications: Synthesis and Surface Functionalization"

_materials, 2021, doi:10.3390/ma14081934_

Round 1

Reviewer 1 Report

A2BC-Type Porphyrin SAM was investigated on the gold surface for the purpose of whole bacterial cell detection. However, some issues need addressing before possible publication.

  1. The advantage of porphyrin in the development of bacteria sensing should be introduced clearly in the part of Introduction. Is it just a bio-linker candidate? What will it improve in the FET-biosensor?
  2. The cyclic voltammetry was used to analyze their electron transfer efficiency and surface coverage after SAM modification, which showed that the porphyrin 5 and porphyrin 10 SAM was fromed on the gold surface. But why were the behaviors of Porphyrin 11 and 12 modified electrodes different? It seems that the peak current increased after the modification of Porphyrin 11 and 12. It should be explained.

Author Response

Dear Reviewer,

thank you for your comments. We kindly accept your suggestions.

1. The advantage of porphyrin in the development of bacteria sensing should be introduced clearly in the part of Introduction. Is it just a bio-linker candidate? What will it improve in the FET-biosensor?

Line 47 – 59: “These porphyrin SAMs have the possibility to act as a bacteria trapping antenna in biosensing systems and might result in high sensitivity and a fast bacteria detection in aqueous solution without further sample preparation.”

2. The cyclic voltammetry was used to analyze their electron transfer efficiency and surface coverage after SAM modification, which showed that the porphyrin 5 and porphyrin 10 SAM was fromed on the gold surface. But why were the behaviors of Porphyrin 11 and 12 modified electrodes different? It seems that the peak current increased after the modification of Porphyrin 11 and 12. It should be explained.

Line 340 – 398: We performed the CV measurements with a new experimental setup, the new results and also the explanation for the increasing peak current are discussed.

Reviewer 2 Report

In this study, authors developed the functionalized porphyrin self-assembled monolayers (SAMs) to be applied for bacteria detection. However, there exists some questions in this manuscript.

1) In this study, authors only investigated the immobilization of functionalized porphyrin, and they did not show anything related to FET biosensor. In my opinion, authors should modify the Title by removing the sensor application.

2) Similar to the previous point, I don't understand why the authors talk a lot about Bacteria biosensors in the introduction section. Their achievement in this study was only investigating the adhesion of functionalized porphyrin self-assembled monolayers (SAMs) instead of any biosensing application. I suggest to re-write the introduction part completely to emphasize the porphyrin modification and adhesion which were the results of this study.

3) In introduction, authors insisted that nucleic acids or exotoxin detection from bacteria require the sample preparation process with long time and cost consumption. However, SAM preparation in this study also require long time and complex processes, even without any sensor preparation. I could not find the novelty or significant advantages of this system for biosensing application.

4) In addition to the results shown in this study, to verify the uniform monolayer formation, I recommend to conduct the surface morphology investigation like atomic force microscopy, and surface plasmon resonance or ellipsometry.

5) In cyclic voltammograms, in the case of porphyrin 11, why does that signal higher than the gold reference electrode? It was not similar to the gold electrode, instead it was obviously higher than that. I recommend to show the reproducibility results of electrochemical investigation with Ipc or Ipa values.

6) In conclusion, authors insisted that this system will be applied for bacteria sensor. Then, authors should at least show the binding of the porphyrin layer to the bacteria. Otherwise, I think that this manuscript should be re-written totally from a different perspective by removal of all the biosensor-related stories. 

7) There exist lots of grammatical errors and mistakes. English editing should be required to modify the grammatical errors and mistakes.

Author Response

Dear Reviewer,

thank you for your comments. We kindly accept your suggestions.

1) In this study, authors only investigated the immobilization of functionalized porphyrin, and they did not show anything related to FET biosensor. In my opinion, authors should modify the Title by removing the sensor application.

Line 2-3: Title changed to “A2BC-Type Porphyrin SAM on Gold Surface for Bacteria Detection Applications: Synthesis and Surface Functionalization”

2) Similar to the previous point, I don't understand why the authors talk a lot about Bacteria biosensors in the introduction section. Their achievement in this study was only investigating the adhesion of functionalized porphyrin self-assembled monolayers (SAMs) instead of any biosensing application. I suggest to re-write the introduction part completely to emphasize the porphyrin modification and adhesion which were the results of this study.

Line 426 – 448: We performed additional experiments to show the E. coli cell immobilization on the functionalized gold surface.

3) In introduction, authors insisted that nucleic acids or exotoxin detection from bacteria require the sample preparation process with long time and cost consumption. However, SAM preparation in this study also require long time and complex processes, even without any sensor preparation. I could not find the novelty or significant advantages of this system for biosensing application.

Line 47 – 59: “These porphyrin SAMs have the possibility to act as a bacteria trapping antenna in biosensing systems and might result in high sensitivity and a fast bacteria detection in aqueous solution without further sample preparation.” Our idea is to develop a sensor which is able to detect the whole bacterial cell within an aqueous solution without further sample preparation, which makes it easier and time saving for the user.

4) In addition to the results shown in this study, to verify the uniform monolayer formation, I recommend to conduct the surface morphology investigation like atomic force microscopy, and surface plasmon resonance or ellipsometry.

We know that the surfaces are nearly uniform covered with the SAMs. Therefore, we think an ellipsometry measurement will lead to no interpretable values. Additionally in some cases the substrates on which the SAMs are connected are semitransparent and therefore not good usable. Some done experiments with our substrates confirmed this hypothesis. AFM measurements of such SAM coated substrates are nearly impossible with our AFM because the resolution was too weak to illustrate single molecule layers.

5) In cyclic voltammograms, in the case of porphyrin 11, why does that signal higher than the gold reference electrode? It was not similar to the gold electrode, instead it was obviously higher than that. I recommend to show the reproducibility results of electrochemical investigation with Ipc or Ipa values.

Line 340 – 398: We performed the CV measurements with a new experimental setup, the new results and also the explanation for the increasing peak current are discussed.

6) In conclusion, authors insisted that this system will be applied for bacteria sensor. Then, authors should at least show the binding of the porphyrin layer to the bacteria. Otherwise, I think that this manuscript should be re-written totally from a different perspective by removal of all the biosensor-related stories. 

Line 426 – 448: We performed additional experiments to show the E. coli cell immobilization on the functionalized gold surface.

7) There exist lots of grammatical errors and mistakes. English editing should be required to modify the grammatical errors and mistakes.

Editing of grammatical errors and mistakes in lines:

Line 14 could → can

Line 18 enables → allows

Line 28 no sufficient → insufficient

Line 29 consequently → which make

Line 59 and proven → and which was proven

Line 109 which were washed before → which were previously washed

Line 121 for 3 days, following → for 3 days, followed

Line 132 afterwards → afterward

Line 132 process was repeated, finally… → process was repeated. Finally,…..

Line 184 with exclusion → under the exclusion of light

Line 185, 199, 223 dest. → dist.

Line 255 Through → Following

Line 262 result → yield

Line 262 It was already shown.. → It has already been..

Line 320 – 321 This gives information about the… → This provides information on the..

Line 334 Another reason could be.. → Another possible reason is,…

Reviewer 3 Report

The presented work deals with very important and current issue of bacteria sensing, and the Authors highlighted this issue on purpose in the Introduction section. Thus, the aim of their experiments is ambitious, however, the Authors failed to prove that A2BC-Type Porphyrin SAMs are really useful for Gram-negative bacteria linkage. The synthesis and physicochemical characterization of SAMs is decent but manuscript lacks the relevant conclusion supported with experimental data.

It would be beneficial for the article quality to perform and describe an experiment on modified porphyrin SAM binding to E.coli cells or even to some components of bacteria cell wall. I suggest to complete the study with this simple experiment to show also the differences in bacteria binding between differently functionalized layers, e.g. how would the cysteine residue in porphyrin 12 affect the binding affinity?

Author Response

Dear Reviewer,

thank you for your comments.

The presented work deals with very important and current issue of bacteria sensing, and the Authors highlighted this issue on purpose in the Introduction section. Thus, the aim of their experiments is ambitious, however, the Authors failed to prove that A2BC-Type Porphyrin SAMs are really useful for Gram-negative bacteria linkage. The synthesis and physicochemical characterization of SAMs is decent but manuscript lacks the relevant conclusion supported with experimental data.

It would be beneficial for the article quality to perform and describe an experiment on modified porphyrin SAM binding to E.coli cells or even to some components of bacteria cell wall. I suggest to complete the study with this simple experiment to show also the differences in bacteria binding between differently functionalized layers, e.g. how would the cysteine residue in porphyrin 12 affect the binding affinity?

Line 426 – 448: Additional experiments for binding of E. coli cells on the porphyrin functionalized gold surface are shown in part 3.3 Fast Lifetime Imaging Microscopy (FLIM)

Round 2

Reviewer 2 Report

In this study, authors developed the functionalized porphyrin self-assembled monolayers (SAMs) to be applied for bacteria detection. However, there exists some questions in this revised manuscript.

1) We know that the surfaces are nearly uniform covered with the SAMs. Therefore, we think an ellipsometry measurement will lead to no interpretable values. Additionally in some cases the substrates on which the SAMs are connected are semitransparent and therefore not good usable. Some done experiments with our substrates confirmed this hypothesis. AFM measurements of such SAM coated substrates are nearly impossible with our AFM because the resolution was too weak to illustrate single molecule layers.

à Relation to author’s response: In my experience, the SAMs could form un-intended region without uniform layer formation sometimes. I still strongly recommend that authors should prove the uniform layer formation for considering the publication. Despite the author's excuse, in my opinion, authors can conduct the SEM and EDS or the other surface investigation techniques such as SPR using the non-transparent gold surfaces. Or authors should provide the other results or data to support the uniform layer formation.

2) We performed the CV measurements with a new experimental setup, the new results and also the explanation for the increasing peak current are discussed.

à Relation to author’s response: In the first review, I also recommend to show the reproducibility results of electrochemical investigation with Ipc or Ipa values. However, I couldn’t find any results related to this. I still recommend to verify the reliability and reproducibility of proposed system.

Author Response

Dear Reviewer,

thank you for your comments.

Response to 1)

We kindly appreciate your advice in performing additional experiments like SEM, EDS or SPR for a proof of the uniform layer formation and we are thankful for your numerous suggestions.   For our research we had the interest to synthesize a porphyrin structure, that is able for a chemisorption on a gold surface and a simultaneously trapping of Gram-negative bacteria. We could show with the help of UV/Vis absorption spectroscopy, CV measurements, contact angle measurements and IRRAS a successful linkage to the surface. With the additional FLIM and bacteria experiments we could also show a linkage of the bacterial cells on the surface, although there might be defects on the surface. The successful linkage of the bacteria cells is the most important prerequisite for our sensor experiments. Of course, we are looking for additional experiments for a proof of the uniform layer formation. But as we don´t have the possibilities in our facility so far, we can´t perform your mentioned experiments in the near future. I hope that you can understand our situation and our point of view.

We added three sentences in our manuscript to make it clear, that we focused on the successful porphyrin linkage to the surface for bacteria trapping:

Line 302 – 303: “Our UV/Vis study shows, that the porphyrins are demonstrably on the surface.”

Line 422 – 424: “With the IRRAS measurements we could also demonstrate, besides the UV/Vis, contact angle und CV measurements, the presence of the porphyrins on the gold surface.”

Line 467 – 469: “As we can successfully connect the Gram-negative bacteria through the porphyrin on the gold surface, these structures are a promising candidate for bacteria sensing applications.”

Response to 2)

We apologize that we have not comment your annotation about the reproducibility of our electrochemical investigations. The measured current during cyclic voltammetry is a superposition of a faradaic current and a diffusion caused mass transfer. If the applied potential exceeds the formal potential the current is dominated by the faradaic current. The surface concentration of the analyte decreases with increasing potential. As the surface concentration drops nearly to zero the peak potential is reached. A further increase of the applied potential leads to an enlargement of the depletion region from the electrode surface to the bulk. The current is then dominated by diffusion of the analyte to the electrode surface and causes the decrease. [S1, S2] The measured current during cyclic voltammetry reflects primary the heterogeneouse one electron transfer between the analyte potassium hexacyanoferrate (II/III) and the gold electrode which can be theoretically determined by the BUTLER-VOLMER-equation.[S1]

Reproducibility in contrast is the verification of the shown results being independent of external influences which is not reflected in one measured value. The used measurement setup is published in different publications and proved the reliability of the used setup [S3-S5]. Maybe the added comment that we have performed cyclic voltammetry measurements with a new experimental setup was misleading. We would like to apologize the confusion. We orientated on many publications before performing cyclic voltammetry in a similar manner [S3-S5] and documented comparable results which are discussed in detail in the revised text passage. Differences in the peak currents and peak potentials are analyzed in consideration of the functional layer. The paper focuses on the verification of self-assembled monolayers with the synthesized porphyrins which could be shown with i. a. cyclic voltammetry unambiguously and should be taken into account in the revision process.

S1.       Bard, A. J.; Faulkner, L. R. Electrochemical Methods, 2nd ed.; Wiley: Danvers, USA, 2001; pp. 226–228.

S2.       Elgrishi, N.; Rountree, K. J.; McCarthy, B. D.; Rountree, E. S.; Eisenhart, T. T. and Dempsey, J. L. A Practical Beginner's Guide to Cyclic Voltammetry, J. Chem. Educ. 2017, 95, 197-206.

S3.       Zuo, G.; Liu, X.; Yang, J.; Li, X. and Lu, X. Study of the adsorption kinetics of thiol-derivatized porphyrin on the surface of gold electrode, Journal of Electroanalytical Chemistry 2007, 605, 81–88.

S4.       Lu, X.; Li, M.; Yang, C.; Zhang, L.; Fi, Y.; Jiang, L.; Li, H.; Jiang, L.; Liu, C.; Hu, W.; Electron transport through a self-assembled monolayer of thiol-end-functionalized tetraphenylporphines and metal tetraphenylporphines, Langmuir 2006, 22, 3035–3039.

S5.       Fischer, L. M.; Tenje, M.; Heiskanen, A. R.; Masuda, N.; Castillo, J.; Bentien, A.; Émneus, J.; Jakobsen, M. H. and Boisen, A. Gold cleaning methods for electrochemical detection applications. Microelectronic Engineering 2009, 86, 1282–1285.

Reviewer 3 Report

In my opinion, the additional experiment is beneficial for the study quality as it proves the thesis highligthed in the title. The manuscript can be accepted in the present form.

Author Response

Thank you very much to accept the manuscript in the present form.

Round 3

Reviewer 2 Report

In this study, authors developed the functionalized porphyrin self-assembled monolayers (SAMs) to be applied for bacteria detection. By following the reviewer’s suggestion, authors modified the ambiguous parts and added the additional explanation well in this revised version.